# Data Hiding Method for Color AMBTC Compressed Images Using Color Difference

**Cheonshik Kim** [1,*] , **Dongkyoo Shin** [1] , **Chingnung Yang** [2] **and Lu Leng** [3,*]

1   Department of Computer Engineering, Sejong University, Seoul 05006, Korea; shindk@sejong.ac.kr
2   Department of Computer Science and Information Engineering, National Dong Hwa University, Hualien 97401, Taiwan; cnyang@gms.ndhu.edu.tw
3   Key Laboratory of Jiangxi Province for Image Processing and Pattern Recognition, Nanchang Hangkong University, Nanchang 330063, China
*   Correspondence: mipsan@sejong.ac.kr (C.K.); leng@nchu.edu.cn (L.L.)

**Abstract:** Image compression technology and copyright protection are certainly the important technologies for free exchange of multimedia. For compression of an image, we propose a color Absolute Moment Block Trucation Coding (AMBTC) method using a common bit-plane created by k-means. In addition, a data hiding method based on a color AMBTC using Optimal Pixel Adjustment Process (OPAP) was proposed for copyright protection and confidential secret communication. The number of quantization levels of the proposed color AMBTC is nine per block. Therefore, the edge of the compressed color image can be expressed more delicately. As a result of the simulation, it can be seen that the edge of the image of the color AMBTC is close to the original image. Moreover, the data hiding performance of the proposed method also obtained excellent results. For the experiment, we measured the quality of the image using the Color Difference (CD) we proposed, and the measurement result was very satisfactory.

**Keywords:** Block Truncation Coding (BTC); Absolute Moment BTC (AMBTC); color AMBTC; data hiding; color difference (CD)

## 1. Introduction

Recently, many people participated in the production and distribution of various media contents with smartphones. Therefore, interest in digital content ownership is increasing more than ever. On the other hand, advanced image processing tools and experienced designers can make the authenticity of digital images difficult [1]. Therefore, "to see is to believe" has become meaningless. Nevertheless, data hiding (DH) [2] is emerging as a potential solution.

DH is used for intellectual property protection, content authentication, annotation, watermarking technology, etc., and supports confidential communication. The difference between DH and watermarking technology is that watermarking technology reveals that digital contents are applied with watermarking technology, but DH technology protects digital contents by hiding the existence of secret information.

DH uses mostly the spatial domain. Thus, data can be efficiently embedded by flipping the redundant bits of the Least Significant Bit (LSB) [3] of the pixels. The merit of spatial domain-based DH [4] is that it is possible to keep a marked high-quality image and to embed enough data. The weakness is that if the LSBs change due to simple image processing, the hidden data in the LSBs can be destroyed. While, although the merit of frequency domain-based DH [5] is that it is strong against image processing attacks, it has the demerit that the Embedding Capacity (EC) is low and the quality of a marked image is relatively not high. Frequency domain compression methods include Discrete Cosine Transform (DCT) [6,7], Discrete Wavelet Transform (DWT) [8], and Singular Value Decomposition (SVD) [9].

Also, BTC (Block Truncation Coding) [10] is being actively researched as a cover media for DH. BTC is a compression coding with fast compression calculation speed and moderate bit rate. Absolute Moment BTC (AMBTC) [11], proposed by Lema and Mitchell, is the most used BTC method. AMBTC quantizes pixels in blocks. As a result of quantization, two pixels and a bitmap representing each block are obtained. This is denoted by $(a, b, BM)$ called trio.

First, the bitmap is directly replaced with the secret bits [12], where a secret bit is a one-to-one relationship with a pixel. This method may be applied if the difference between the two quantization levels of blocks is less than $T$ (threshold value) ($b - a \leq T$). Here, DH performance may vary depending on the texture of the cover image. Second, there is a lossless DH [13] using the order of two quantization levels (OTQL). For example, to embed the '1' bit, the order of $a$ and $b$ is reversed as follows: $trio(b, a, BM)$. Since this method does not change the coefficients of the two quantization levels, the original cover image can be completely restored. Third, Ou & Sun [14] introduces a method that combines DBR (Directly the Bitmap Replacement) and OTQL. Moreover, they introduced an optimization equation that reduces errors generated when bit-planes are changed. Fourth, Huang et al. [15] introduced a method embedding as much as $log_2^T$ bits by using difference expansion of two quantization levels. The merit of this method does not depend on the type of blocks. Fifth, Kim et al. [16] proposed an effective DH method for two quantization levels of each block of AMBTC using Hamming code. In order to solve the image distortion error problem, they introduced a method of optimizing the codeword and reducing pixel distortion by using a lookup table.

Meanwhile, since a color image consists of R, G, and B layers, it is compressed into six quantization levels and three bitmaps per block and can be represented as 96 (=6 × 8 + 16 × 3) bits per block (e.g., the 4 × 4 sized block). On the other hand, our proposed color AMBTC is represented with 9 quantization levels and a common bit-plane per block. Instead of increasing the number of quantization levels per block, one bit per block can be compressed more than the conventional ABMTC by using a common bitmap and 7-bit quantization levels. In other words, the image can be represented with about 95 bits per block. Specifically, to compress three bitmaps into one bitmap, we extract the luminance from RGB and use k-means [17] to classify the pixels in the block into three groups to generate a common bitmap per block. The performance of this method can be confirmed by experimental results.

In this paper, we introduce a method for efficient DH based on our proposed color AMBTC. For efficient DH, we use the Optimal Pixel Adjustment Process (OPAP) method [18]. Through the experiment, we can confirm the superiority of the color AMBTC based on k-means. The contributions we proposed to color ABMTC and DH methods are as follows. First, the color AMBTC method using k-means enables improving the image quality dramatically. Second, the image compression performance was improved by 1 bit per block. Third, we showed how to use OPAP to hide enough secret data while reducing the loss of the cover image.

The rest of this paper is organized as follows: Section 2 reviews the BTC series used to compress gray images and the BTC series related to color image compression. The encoding process of color AMBTC and generation of the common bit-plane are discussed in Section 3. In Section 4, the proposed color AMBTC method and the data hiding method based on it are introduced. Section 5 are the experimental result. Section 6 draws some conclusions.

## 2. Related Works

In this section, we survey the evolution of the BTC-based compression methods, the color image compression methods.

A lossy compression technique commonly known as Block Truncation Coding (BTC) [10] was introduced in 1979 by Delp and Mitchell. The BTC first divides the pixels of an original image into sized 4 × 4 non-overlapping blocks, then calculates the mean and standard deviation of each block, and then if the pixels in each block are greater than the mean, '1' is assigned, otherwise, '0' is assigned to generate the bit-plane. Finally, two quantization

levels representing each block are generated using the mean and standard deviation. BTC provides a relatively high compression performance while the calculation is simple. Since BTC, the method that provides stable performance is Absolute Moment BTC (AMBTC) [11], which was proposed in 1984 by Lema & Mitchell.

In 1992, Wu and Coll [19] discussed a new block coding for color images, where they introduced a method of constructing a common bit-plane using three rules (majority, luminance, weight). The bit rate was improved using this method. Encoded BTC is further compressed using adaptive DCT coding. Differential coding and residual error feedback techniques have been introduced to reduce the bit rate and improve mean squared error (MSE) [20] performance. The total compression ratio is about 12:1.

Color image compression moment BTC (CICMPBTC) [21] was proposed by Yang to improve the compression ratio of color AMBTC (CAMBTC) using a traditional method. After generating BTC, the spectral identification code is used to record the quantization level information. This method consumes more time in the compression process than CAMBTC and causes a significant loss in image quality during the compression process.

Chang et al. [22] discussed the Two-Layer AMBTC scheme (TLAMBTC) to compress color images. Each quantization level is obtained from each of the CAMBTC-based R, G, and B layers divided into blocks. Then, a common bit-plane is derived by using the correlation. The next step is to code the quantization level by subdividing it into $2 \times 2$ blocks. Hu et al. [23] proposed a color image compression method combining block predictive coding and AMBTC, where quantization level coding are for use to compress blocks, respectively. The bit-plane is generated by using appropriate weights to the corresponding R, G, and B components. This method gives a contribution to the MSE performance of the compressed image.

The problem is that some color components on one plane may not correlate with other planes. Thus, choosing one of the R, G, or B bitmaps is not appropriate and can cause serious noise. To improve this problem, we extract the luminance from each block (composed of RGB) and apply k-means to the luminance (grayscale) of each block. With this, the common bit plane is extracted and used.

### 3. Preliminaries

*3.1. Color AMBTC*

AMBTC [11] is a lossy compression method for grayscale images and preserves moment quantization values based on blocks. After dividing the image by sized $m \times m$ blocks, it is compressing by creating two quantization levels and a bitmap for each block.

This is due to the similarity of pixels within the block. However, as much as the difference between the compressed pixels and the original pixels causes the distortion of the compressed image, but the difference is relatively small. If the block size is $4 \times 4$, the bit rate is 2 bits per pixel for the block because compression rate (CR) (=size of original image/size of compressed image) is 4. The merits of AMBTC are simple computation and fast image compression.

The AMBTC encoding is performed as follows.

Step 1: The original image of size $n \times n$ is divided into non-overlapping blocks ($P$) of size $m \times m$ (let $m = 4$) and each block is processed individually. Also let us say $m^2 = k$.

Step 2: For each block $P_i$, the mean of the pixels constituting $P_i$ is calculated and assigned to $\bar{x}$ (Equation (1)), where the variable $i$ denotes $\{1 \leq i \leq n \times n/(m \times m)\}$ and $j$ denotes $\{1 \leq j \leq m \times m\}$.

$$\bar{x} = \frac{\sum_{j=1}^{k} x_j}{k} \tag{1}$$

Step 3: A bit-plane for the block $P_i$ is caused by a simple calculation of Equation (2), i.e., if the pixel $x_j$ is greater than equal to the mean ($\bar{x}$) then it is assigned a '1', otherwise it is assigned a '0'. As a result, a bitmap $M$ consisting of 0s and 1s is obtained.

$$b_j = \begin{cases} 1, & \text{if } x_j \geq \bar{x}, \\ 0, & \text{if } x_j < \bar{x}. \end{cases} \tag{2}$$

Step 4: The bitmap $M$ is divided into two sets, $M_0$ and $M_1$, where $M = M_0 \cup M_1$ and $M_0 \cap M_1 = \phi$. $M_0$ and $M_1$ are $M_0 = \{0_0, 0_1, \ldots, 0_t\}$, $M_1 = \{1_0, 1_1, \ldots, 1_{(k-t)}\}$, respectively. Two values of $t$ and $k - t$ are each number of pixels of group '0' and '1'. The means of pixels $Q_1$ and $Q_2$ in the two groups represent the quantization levels of groups '0' and '1', respectively. The two quantization levels are calculated by Equations (3) and (4).

$$Q_1 = \left\lfloor \frac{1}{t} \sum_{x_j < \bar{x}} x_j \right\rfloor \tag{3}$$

$$Q_2 = \left\lfloor \frac{1}{k-t} \sum_{x_j \geq \bar{x}} x_j \right\rfloor \tag{4}$$

Step 5: Two quantization levels $Q_1$ and $Q_2$ and a bitmap $M$ are added to a *trio*, i.e., *trio* $\leftarrow$ *trio* $|| \{Q_1, Q_2, M\}$.

Step 6: Steps 2 to 5 are repeated until encoding is completed for all blocks.

The two quantization levels $Q_1$ and $Q_2$ can be used in the process of reconstructing the compressed image, and the method by using Equation (5), where $g_j$ denotes a pixel of grayscale.

$$g_j = \begin{cases} Q_1, & \text{if } b_j = 0, \\ Q_2, & \text{if } b_j = 1. \end{cases} \tag{5}$$

To compress a color image using traditional AMBTC (CAMBTC), the RGB, which is the color image as shown in Figure 1, is decomposed into layers R, G, and B, and then the AMBTC method is applied to each layer. A block of CAMBTC format is $trio\_RGB_i = \{Q_1^R, Q_2^R, M_1^R, Q_1^G, Q_2^G, M_2^G, Q_1^B, Q_2^B, M_3^B\}$. The compression code of each RGB color image block consists of 3 planes and 6 quantization levels.

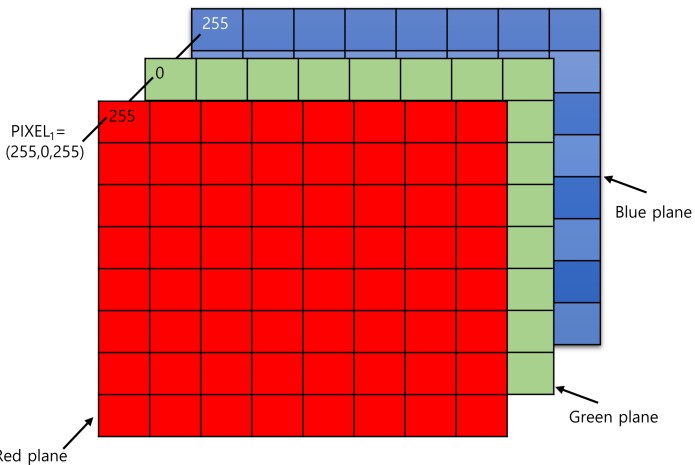

**Figure 1.** RGB image consisting of three layers.

*3.2. Optimal Pixel Adjustment Process (OPAP)*

LSB substitution method is a DH by directly replacing the LSBs of the cover image according to the secret bits. Wang et al. [18] introduced DH, which allows for optimal LSB substitution, where the worst mean square error (WMSE) was found to be 1/2 that obtained with the simple LSB substitution method. We assume that a pixel in a grayscale 8-bit is $x_i \in \{0, 1, \ldots, 255\}$ and $\mathcal{S}$ means $n$-bit secret bits as $\mathcal{S} = \{s_k | 0 \leq k < n, s_k \in \{0, 1\}\}$. The relationship between the $\mathcal{S}$ and embedded bits $\mathcal{S}'$ is defined as $s_k' = \sum_{j=0}^{\delta-1} s_k \times \delta + j \times 2^{\delta-1-j}$.

For embedding the $\delta$-bit, the pixel $x_i$ is changed to be $x_i' = x_i - (x_i \bmod 2^\delta) + s_k'$. The hidden $\delta$ LSBs of $x_i$ are extracted by using $s_k = x_i' \bmod 2^\delta$. Let $x_i$ and $x_i'$ be the pixels of the cover image and stego image, respectively. The pixel $x_i''$ denotes the optimized pixel derived from $x_i'$ by using OPAP method. Let $\Delta_i = x_i' - x_i$ be the embdding error between $x_i$ and $x_i'$. The pixel $x'$ optimizes as $x''$ as the following rules:

1.  $(2^{\delta-1} < \Delta < 2^\delta)$: if $x_i' \geq 2^\delta$, then $x_i'' = x_i' - 2^\delta$; else $x_i'' = x_i'$;
2.  $(-2^{\delta-1} \leq \Delta \leq 2^{\delta-1})$: $x_i'' = x_i'$;
3.  $(-2^\delta < \Delta_i < -2^{\delta-1})$: if $x_i' < 256 - 2^\delta$, then $x_i'' = x_i' + 2^\delta$; else $x_i'' = x_i'$.

## 4. Proposed Method

In this section, we propose a color AMBTC using k-means and introduce a DH and extraction methods based on it. Finally, we explain the proposed methods with examples easy to understand.

*4.1. The Proposed Color AMBTC Using k-Means*

When a color image is compressed with traditional color AMBTC, three bit-planes are used per block, but the method suggested in this section uses a common bit-plane to improve compression performance. The common bit plane is obtained by the k-mean clustering. The detailed compression process is described in stages.

Input: Original image, $\mathcal{OI}$; The image size, $N \times N$

Output: Color AMBTC compressed image, $\mathcal{CI}$

Step 1: The original image sized $n \times n$ is divided into non-overlapping blocks ($\mathcal{P}$) of size $m \times m$ (let $m$ = 4) and each block is processed individually. Also let us say $m^2 = k$. Here, the variable $i$ denotes $\{1 \leq i \leq n \times n/(m \times m)\}$ and $j$ denotes $\{1 \leq j \leq m \times m\}$.

Step 2: For each block $\mathcal{P}_i$, transform a block of RGB layer into a common block containing the luminance using Equation (6), where $p_j$ is a pixel.

$$p_j' = 0.299 \times p_j^R + 0.587 \times p_j^G + 0.114 \times p_j^B \tag{6}$$

Step 3: Apply the k-means clustering algorithm to the pixels of $\mathcal{P}_i'$. Here, the number of clusters is k (= 3). After executing the k-means, a common bitmap $M$ composed of {00,01,10} is obtained as in Figure 2.

Step 4: The common bitmap $M$ is divided into three sets: $M_0$, $M_1$ and $M_2$, where $M = M_0 \cup M_1 \cup M_2$ and $M_0 \cap M_1 \cap M_2 = \phi$. Here, $M_0$, $M_1$, and $M_2$ are $M_0 = \{00_0, 00_1, \ldots, 00_{t_0}\}$, $M_1 = \{01_0, 01_1, \ldots, 01_{t_1}\}$, and $M_2 = \{10_0, 10_1, \ldots, 10_{t_2}\}$. $t_o$, $t_1$, and $t_2$ represent the numbers of pixels in each group for '00', '01', and '10'. $Q_1$, $Q_2$, and $Q_3$ denote quantization levels for each group. Using Equations (7)–(9), the quantization levels of R, G, and B are $Q^R(Q_1, Q_2, Q_3)$, $Q^G(Q_1, Q_2, Q_3)$, and $Q^B(Q_1, Q_2, Q_3)$ can be obtained.

In the following equations, the variables such as R, G, and B refer to the R, G, and B layers of block $\mathcal{P}_i$ and each layer consists of $m \times m$ pixels.

$$Q^R_{i_1} = \begin{cases} \lfloor \frac{1}{t_0} \sum_{j=1}^{t_0} R_j \rfloor, & \text{if}(M_j = \text{'00'}), \\ \lfloor \frac{1}{t_1} \sum_{j=1}^{t_1} R_j \rfloor, & \text{if}(M_j = \text{'01'}), \\ \lfloor \frac{1}{t_2} \sum_{j=1}^{t_2} R_j \rfloor, & \text{if}(M_j = \text{'10'}). \end{cases} \tag{7}$$

$$Q^G_{i_2} = \begin{cases} \lfloor \frac{1}{t_0} \sum_{j=1}^{t_0} G_j \rfloor, & \text{if}(M_j = \text{'00'}), \\ \lfloor \frac{1}{t_1} \sum_{j=1}^{t_1} G_j \rfloor, & \text{if}(M_j = \text{'01'}), \\ \lfloor \frac{1}{t_2} \sum_{j=1}^{t_2} G_j \rfloor, & \text{if}(M_j = \text{'10'}). \end{cases} \tag{8}$$

$$Q^B_{i_3} = \begin{cases} \lfloor \frac{1}{t_0} \sum_{j=1}^{t_0} B_j \rfloor, & \text{if}(M_j = \text{'00'}), \\ \lfloor \frac{1}{t_1} \sum_{j=1}^{t_1} B_j \rfloor, & \text{if}(M_j = \text{'01'}), \\ \lfloor \frac{1}{t_2} \sum_{j=1}^{t_2} B_j \rfloor, & \text{if}(M_j = \text{'10'}). \end{cases} \tag{9}$$

Step 5: To compress the quantized values for each of R, G, and B once again, the LSB is compressed in a way that does not use the LSBs for $Q_1$, $Q_2$, and $Q_3$. To do this, the LSBs are removed by Equation (10), where $\lfloor \cdot \rfloor$ is a truncation function.

$$Q_i = \sum_{j=1}^{9} \left\lfloor \frac{Q_j}{2} \right\rfloor \times 2 \tag{10}$$

The expression range of the quantized pixel value $Q$ is $(b_7, b_6, \ldots, b_1) = (128\ 64\ 32\ 16\ 8\ 4\ 2)$.

Step 6: The compressed data is added to the $\mathcal{CI}$, i.e., $\mathcal{CI} \leftarrow \mathcal{CI}\ ||\ \{Q^R_1, Q^R_2, Q^R_3, Q^G_1, Q^G_2, Q^G_3, Q^B_1, Q^B_2, Q^B_3, M, \ldots\}$. Repeat *Steps* 2 to 6 until all blocks are extruded.

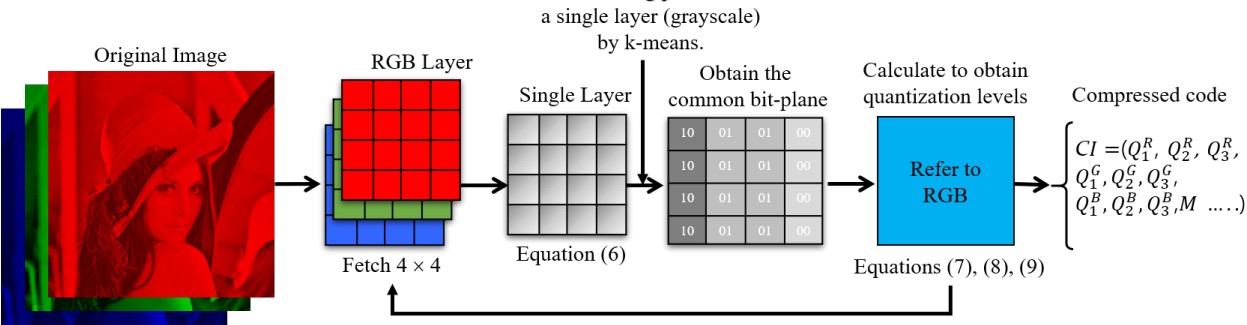

**Figure 2.** The proposed color AMBTC generation process.

After performing the proposed AMBTC, it is compressed to 95 bits per block, and the compression efficiency is improved by one bit more than that (96 bits per block) compared with CAMBTC, introduced in Section 2. Moreover, three quantization levels were used to consider the quality of the image, and for compression, the expression range of each quantization level was proposed to be seven bits. Therefore, the total compression bits for one block (4 × 4) in the proposed color AMBTC-based image is $32 \times 32 + (7 + 7 + 7) \times 3 = 95$, where 32 denotes the number of bits of a common bitmap.

### 4.2. DH Embedding Procedure

In Section 4.1, we introduced a method of compressing the original image with the proposed color AMBTC encoding method. After performing the AMBTC, it was shown that compression bits are 95 per block. In this section, we propose a method to embed

secret bits while minimizing distortion using OPAP. The secret bits are embedded in the compressed $\mathcal{CI}$ depending on the procedure in Figure 3. The blocks in $\mathcal{CI}$ are composed of a common bit-plane and three quantization levels for R, G, and B, respectively. Nine quantization levels and a common bit-plane may embed 18 bits ($9 \times 2$ bits) and 16 bits per block, respectively. The detailed DH embedding procedure is described in stages.

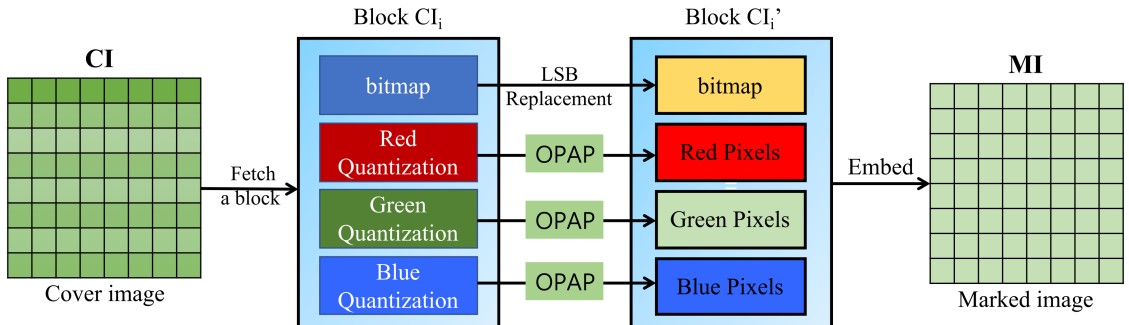

**Figure 3.** Schematic diagram of DH structure.

Input: Color AMBTC compressed image, $\mathcal{CI}$; block size, $n \times n/(m \times m)$; secret bits, $\mathcal{S}$

Output: Color AMBTC marked image, $\mathcal{MI}$

Step 1: Read a block from $\mathcal{CI}$ and assign it to $\mathcal{P}_i$, where $i \in \{1 \le i \le (n \times n)/(m \times m)\}$.

Step 2: The quantization levels of block $\mathcal{P}_i$ is assigned to the variable $Q_i$, where the range of $j$ is $\{1 \le j \le 9\}$ and $Q_j \in \{0, 1, \ldots, 255\}$. Thereafter, LSB and 2LSB are extracted from $Q_j$ using Equation (12). Next, the secret bit $S_k$(1 bit) is concealed in the 2LSB of $Q_j$ according to the rule of Equation (13), i.e., $\mathcal{P}'_i(j) = \sum_{j=1}^9 f(\mathcal{P}_i, \mathcal{S}_k)$, where $f$ is a function of the embedding logic.

$$Q_j = \sum_{j=1}^9 \mathcal{P}_i(j) \tag{11}$$

$$\begin{cases} b_j^1 \xleftarrow{\text{LSB}} Q_j \bmod 2 \\ b_j^2 \xleftarrow{\text{2LSB}} \lfloor Q_j/2 \rfloor \bmod 2 \end{cases} \tag{12}$$

$$Q'_j = \begin{cases} Q_j - 1, & \text{if}(b_j^2 = 0 \wedge b_j^1 = 0) \wedge S_k = 1 \\ Q_j + 1, & \text{if}(b_j^2 = 0 \wedge b_j^1 = 1) \wedge S_k = 1 \\ Q_j - 1, & \text{if}(b_j^2 = 1 \wedge b_j^1 = 0) \wedge S_k = 0 \\ Q_j + 1, & \text{if}(b_j^2 = 1 \wedge b_j^1 = 1) \wedge S_k = 0 \end{cases} \tag{13}$$

After performing Step 2, nine bits are hidden in the 2LSB of nine quantization levels.

Step 3: Embedding the secret bit $\mathcal{S}_k$ in XOR (2LSB $\oplus$ LSB) of $Q'_j$ depends on the OPAP rule (Equation (14)), where 2LSB $\oplus$ LSB is $(b_j^2 \oplus b_j^1)$. Since the pixel value of $Q'_j$ can be changed by Equation (13), Equations (11) and (12) are recalculated to extract 2LSB and LSB again. The function call for Equation (14) is $\mathcal{P}'_i(j) = \sum_{j=1}^9 f(\mathcal{P}_i, \mathcal{S}_k)$, where $f$ is a function of the embedding logic.

$$\mathcal{P}'_i(j) = \begin{cases} Q'_j - 1, & \text{if}(b_j^2 = 1 \wedge b_j^1 = 1) \wedge S_k = 1 \\ Q'_j + 1, & \text{if}(b_j^2 = 1 \wedge b_j^1 = 0) \wedge S_k = 0 \\ Q'_j - 1, & \text{if}(b_j^2 = 0 \wedge b_j^1 = 1) \wedge S_k = 0 \\ Q'_j + 1, & \text{if}(b_j^2 = 0 \wedge b_j^1 = 0) \wedge S_k = 1 \\ nochange, & otherwise \end{cases} \tag{14}$$

Step 4: A critical condition to embed a stream of secret bits into a common bit-plane is allowed when the difference between two quantization levels is less than $T$. If $(Q_1^G - Q_2^G) \leq T$, read a common bit-plane $M$ from the block $\mathcal{P}_i'$ and perform Equation (15) to embed the secret bits into $M$, i.e., $P_i'(j) = \sum_{j=1}^{32} f(M_j, \mathcal{S}_k)$, where $f$ is a function of the logic.

$$M_j : M_{j+1} = \begin{cases} dec(M_j : M_{j+1}) + 1, & \text{if}(M_{j+1} \neq \mathcal{S}_k \wedge M_j : M_{j+1} = 00|01), \\ dec(M_j : M_{j+1}) - 1, & \text{if}(M_{j+1} \neq \mathcal{S}_k \wedge M_j : M_{j+1} = 10), \\ inc(j+2), inc(k). \end{cases} \quad (15)$$

A threshold value $T$ is a condition for the embedding secret bits in a common bitmap as well as controls the quality of color AMBTC image.

Step 5: Steps 1 to 5 are repeated until all blocks have been processed.

### 4.3. DH Extraction Procedure

We assumed that the marked image $\mathcal{MI}$ is delivered to the receiving side. The recovering procedure hidden secret bits is performed by reversing order of the embedding procedure (Figure 3), i.e., read a block from $\mathcal{MI}$ in order, and extract hidden bits from both of the quantization levels and a common bitmap. The detailed DH extraction procedure is described in stages.

Input: The marked image, $\mathcal{MI}$; block size, $n \times n/(m \times m)$;.

Output: Extracted secret bits, $\mathcal{S}'$

Step 1: Read a block from $\mathcal{MI}$ and assign it to $\mathcal{P}_i$, where $i \in \{1 \leq i \leq (n \times n)/(m \times m)\}$.

Step 2: To extract hidden bits in nine quantization levels, Equation (16) is executed.

$$[S_k] = \sum_{j=1}^{9} \begin{cases} \overset{\text{2LSB}}{\longleftarrow} \left\lfloor \frac{Q_i(j)}{2} \right\rfloor \bmod 2; inc(k), \\ \overset{\text{2LSB} \oplus \text{LSB}}{\longleftarrow} \text{XOR}\left( \left\lfloor \frac{Q_i(j)}{2} \right\rfloor \bmod 2, Q_i(j) \bmod 2 \right); inc(k). \end{cases} \quad (16)$$

Step 3: If $\mathcal{P}_i (Q_1^G - Q_2^G) \leq T$, the common bitmap in block $\mathcal{P}_i$ is assigned to variable $M$. Then, 16-bit data is extracted using Equation (17), where $j$ is $\{1 \leq j \leq 32\}$, i.e., $\mathcal{S}_k' = \sum_{j=1}^{32} f(M_j)$, where $f$ is a function of the logic.

$$[S_k] = \begin{cases} 0, & if(M_j : M_{j+1}) = (00)|(10), \\ 1, & if(M_j : M_{j+1}) = (01), \\ inc(j+2), inc(k). \end{cases} \quad (17)$$

Step 4: To extract hidden data from all blocks, the process of *Steps* 1 to 3 is repeated.

### 4.4. AMBTC Compression and DH Example

**Example 1.** *Traditional color AMBTC method.*

It denotes $\mathcal{OI}_R, \mathcal{OI}_G$, and $\mathcal{OI}_B$ as layers of R, G, and B of the original color image $\mathcal{OI}$. We describe the traditional color AMBTC encoding procedure with an example of Figure 4, where a block size is $4 \times 4$.

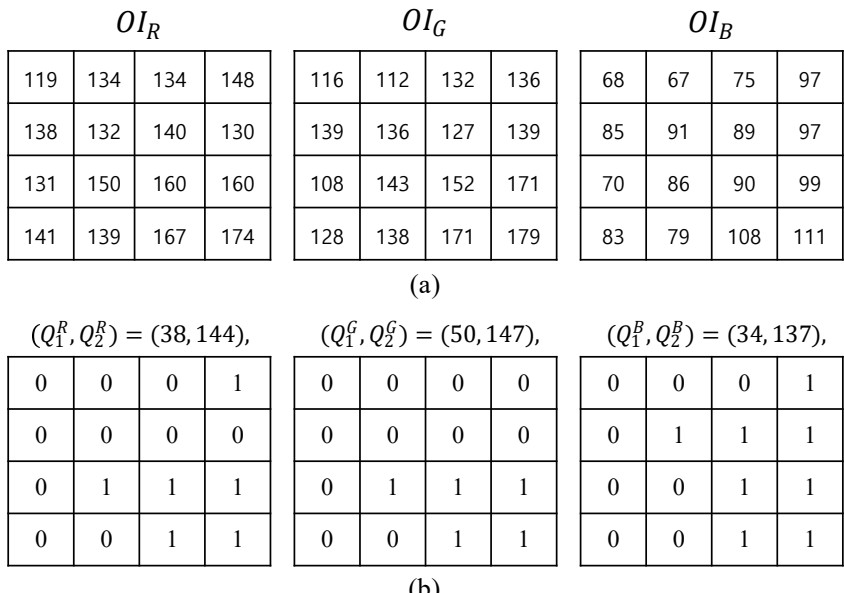

$OI_R$

| 119 | 134 | 134 | 148 |
|---|---|---|---|
| 138 | 132 | 140 | 130 |
| 131 | 150 | 160 | 160 |
| 141 | 139 | 167 | 174 |

$OI_G$

| 116 | 112 | 132 | 136 |
|---|---|---|---|
| 139 | 136 | 127 | 139 |
| 108 | 143 | 152 | 171 |
| 128 | 138 | 171 | 179 |

$OI_B$

| 68 | 67 | 75 | 97 |
|---|---|---|---|
| 85 | 91 | 89 | 97 |
| 70 | 86 | 90 | 99 |
| 83 | 79 | 108 | 111 |

(a)

$(Q_1^R, Q_2^R) = (38, 144),$

| 0 | 0 | 0 | 1 |
|---|---|---|---|
| 0 | 0 | 0 | 0 |
| 0 | 1 | 1 | 1 |
| 0 | 0 | 1 | 1 |

$(Q_1^G, Q_2^G) = (50, 147),$

| 0 | 0 | 0 | 0 |
|---|---|---|---|
| 0 | 0 | 0 | 0 |
| 0 | 1 | 1 | 1 |
| 0 | 0 | 1 | 1 |

$(Q_1^B, Q_2^B) = (34, 137),$

| 0 | 0 | 0 | 1 |
|---|---|---|---|
| 0 | 1 | 1 | 1 |
| 0 | 0 | 1 | 1 |
| 0 | 0 | 1 | 1 |

(b)

**Figure 4.** An example of the procedure of a block of traditional color AMBTC; (**a**) RGB blocks sized $4 \times 4$ of original image $OI$, (**b**) the three bitmaps of color AMBTC $CT_T$.

Traditional color AMBTC is compressed into two quantization levels and one bitmap for each layer of R, G, and B as shown in Figure 4. The mean (144) and bitmap (Figure 4b) based on the $4 \times 4$ of $\mathcal{OI}_R$ can be obtained by using Equations (1) and (2). For $\mathcal{OI}_G$ and $\mathcal{OI}_B$, the mean and a bitmap can be obtained in the same way, respectively. As for the quantization level, $(Q_1^R, Q_2^R) = (38, 144)$ can be obtained as shown in Figure 4b as performing Equations (3) and (4).

**Example 2.** *The proposed color AMBTC method.*

With the RGB blocks in Figure 4a in Example 1, the obtaining procedure of a common bit-plane is explained. First, a block (Figure 5a) of luminance is obtained using Equation (6) and Figure 5b shows this clarification of pixels by using k-means algorithm. Here, The number of clustering is 3. If three quantization levels are calculated based on the bitmap by the procedure of Equations (7)–(9), the quantization level as shown in Figure 5 can be obtained. After further quantization is performed using Equation (10), the following results are obtained.

$$Q^R = (136, 162, 174); Q^G = (128, 64, 178); Q^B = (82, 98, 110);$$

$$M = (`00000000000000000000010100000110')$$

$(Q_1^R, Q_2^R, Q_3^R) = (136, 162, 174); (Q_1^G, Q_2^G, Q_3^G) = (129, 164, 179); (Q_1^B, Q_2^B, Q_3^B) = (82, 99, 111)$

| 111 | 113 | 126 | 135 |
|---|---|---|---|
| 132 | 129 | 126 | 131 |
| 110 | 138 | 147 | 159 |
| 126 | 131 | 162 | 169 |

(a)

| 00 | 00 | 00 | 00 |
|---|---|---|---|
| 00 | 00 | 00 | 00 |
| 00 | 00 | 01 | 01 |
| 00 | 00 | 01 | 10 |

(b)

**Figure 5.** The proposed compress procedure of a block of color AMBTC; (**a**) obtained grayscaled image $GI$, (**b**) bitmap of the proposed AMBTC $MI_P$.

Unlike the quantization levels obtained in Example 1, there are three quantization levels for each color, thus it is possible to represent the medium value for the block as well as it has excellent compression performance.

**Example 3.** *DH procedure using an example.*

To explain the embedding procedure, we use the quantization levels and a bitmap obtained in Example 1. Assuming that secret bit $\mathcal{S}$ = (111000) and threshold $T$ = 60. Moreover, only $Q^R$ is considered for avoiding repeated embedding process for describing this example. First, in order to hide 2-bit data in $Q_1^R = 136$, which is the first pixel of $Q^R = (136, 162, 174)$ and then, transform to $Q_1^{R'} = \lfloor 136/2 \rfloor = 68$. Next, extract 2LSB and LSB, i.e., (2LSB LSB) = (0 0) and $\mathcal{S}_1 = 1$. According to Equation (13), $Q_1^{R''} = Q_1^{R'}(68) - 1 = 67$ is obtained. Since (2LSB LSB) = (1 1) and $S_2 = 1$, we obtain $Q_1^{R''}(67) - 1 = 66$ by using Equation (14). $Q_1^{R''}(66)$ is recovered by the calculation, $Q_1^{R''} \times 2 = 132$. Next, for $Q_2^R$, the way of embedding $S_{3,4}$ in $Q_2^R$ is the same procedure of $Q_1^R$. After embedding $S_{3,4}$ in $Q_2^R$, it is $Q_2^R = 166$. In case of $Q_3^R$, when $S_{5,6} = (00)$ is embedded in $Q_R^3$, it become $Q_R^3 = 176$.

The condition for embedding secret bits in the common bitmap is that the difference between the two quantization levels ($|Q_1^G - Q_2^G| \leq T$) must be less than or equal to the threshold value $T$. Here, it is less than $T = 60$, thus the criterion is enough. Therefore, it can conceal a given 16-bit secret bit. Assuming $S = (0010100101001101)$, the 16 bits are hidden in bitmap $M$ using Equation (15). $S_{7,8}$ is (00), which is the same as $M$, so there is no need to change $M$. For the following $S_{9,10} = (10)$, +1 is performed according to the rule of Equation (15). For embedding the remaining secret bits, if the bitmap $M$ is applied successively by using Equation (15), it becomes $M = (0000010001000001000110100101001)$.

## 5. Experimental Results

In this section, we experiment and evaluate the proposed DH method based on color AMBTC with five test images such as Lena, Pepper, Airplane, Lake, and Baboon (Figure 6). The size of the images used in the experiment is $512 \times 512$ and the size of a block is $4 \times 4$. For evaluation, we use a Color Difference (CD) to compare the performance and use PSNR and SSIM as a reference.

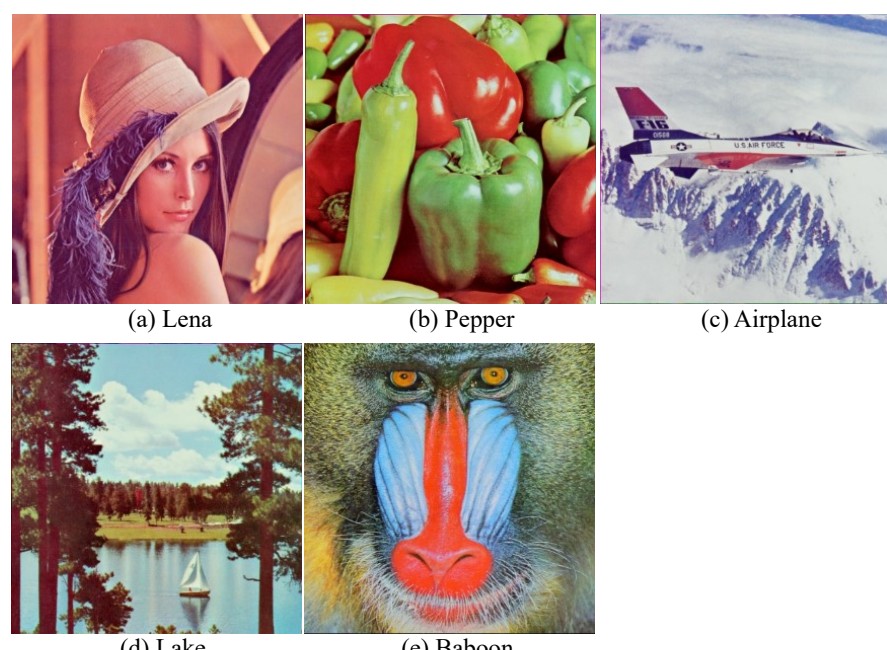

(a) Lena　　　　　　　　　(b) Pepper　　　　　　　　　(c) Airplane

(d) Lake　　　　　　　　　(e) Baboon

**Figure 6.** Original image used in the experiment; (**a**) Lena, (**b**) Pepper, (**c**) Airplane, (**d**) Lake, and (**e**) Baboon.

The quality of the image was measured by the PSNR defined as

$$PSNR = 10 log_{10} \frac{255^2}{(MSE_R + MSE_G + MSE_B)/3} \tag{18}$$

PSNR is calculated as $10 log$ (signal power/noise power), and signal power and noise power are calculated using peak power. The $MSE_R$, $MSE_G$, and $MSE_B$ used for PSNR calculation is the difference in mean intensity between the marked image and the reference image, and a low MSE value can be evaluated as good image quality. In other words, the MSE is the mean of the squares of the errors $(p_i - p'_i)^2$, where $p$ and $p'$ are reference and distorted images, respectively. The MSE is calculated as follows:

$$MSE_{R,G,B}(p, p') = \frac{1}{n \times n} \sum_{i=1}^{n \times n} (p_i - p'_i)^2. \tag{19}$$

Here, the allowable pixel intensity is $255^2$.

SSIM [24] is a multiplicative model for three similarity components i.e., luminance, contrast and structure similarity components, and it is formulated as

$$\phi(p, p') = l(p, p')^{\theta_1} \cdot c(p, p')^{\theta_2} \cdot s(p, p')^{\theta_3} \tag{20}$$

where $l(\cdot)$, $c(\cdot)$ and $s(\cdot)$ indicate luminance, contrast and structure similarity components, respectively, and exponents $\theta_1$, $\theta_2$ and $\theta_3$ are set to 1 in [25].

$l(\cdot)$ in Equation (20) measures similarity of mean pixel intensity between $p$ and $p'$, which is expressed as

$$l(p, p') = \frac{2\mu_p \mu_{p'} + C_1}{\mu_p^2 + \mu_{p'}^2 + C_1} \tag{21}$$

where $\mu_p$ and $\mu_{p'}$ are mean pixel intensity for $p$ and $p'$, respectively, and $C_1$ is a parameter to avoid unstable results when the denominator in Equation (21) is close to zero. The overall measured quality value of the SSIM for the whole distorted image $P'$ compared to the original image $P$ is calculated as

$$F(P, P') = \frac{1}{n \times n} \sum_{j=1}^{n \times n} \phi(p_j, p'_j) \tag{22}$$

where $n \times n$ is the number of local image regions of an whole image and $j$ is an index of local image regions, $p_j$ and $p'_j$.

The quality measurement of the color image we proposed is by the Color Difference (CD) defined as

$$CD(p, p') = \sum_{i=1}^{n \times n} \mathcal{D}(p_i) - \mathcal{D}(p'_i). \tag{23}$$

CD measures similarity by color difference. In case of the measurement of CD, it is the use of weighted R, G, and B values to better fit human perception. A color difference in each pixel is calculated by

$$D(x) = \sqrt{2 \times 0.3x_R^2 + 4 \times 0.59x_G^2 + 3 \times 0.11x_B^2}. \tag{24}$$

where the R, G, and B components are weighted by 30%, 59%, and 11%, respectively. Unlike the other measurement methods, the CD can differentiate and compare feelings of the images' surface and textures that humans can perceive by the visual system.

Table 1 shows the comparison of compression performance and image quality of traditional AMBTC and our proposed AMBTC methods by using CD, SSIM, and PSNR. From an image compression point of view, when compressing with traditional AMBTC, it compresses at 96 bits per block. On the contrary, our proposed method can compress at about 95 bits per block. Here, 2Q/B means that two quantization levels are used per block, and 3Q/B means that three quantization levels are used per block. Our proposed method compresses to 7 bits for each of three quantization levels.

**Table 1.** Comparison of the quality between the tradition color AMBTC and the proposed AMBTC.

| | CAMBTC (96 Bits) 2Q/B | | | PROPOSED (95 Bits) 3Q/B | | |
|---|---|---|---|---|---|---|
| | $CD_t(O-M)$ | SSIM | PSNR | $CD_p(O-M)$ | SSIM | PSNR |
| Lena | 11.5419 | 0.9887 | 37.8912 | 10.6977 | 0.9914 | 39.6064 |
| Pepper | 12.4374 | 0.9882 | 37.4795 | 12.2976 | 0.9891 | 38.476 |
| Airplane | 10.3398 | 0.9346 | 37.1915 | 8.5637 | 0.9312 | 40.4584 |
| Lake | 18.4897 | 0.9618 | 34.0499 | 17.0456 | 0.9623 | 35.316 |
| Baboon | 29.7421 | 0.9253 | 31.1472 | 26.1272 | 0.9322 | 32.2806 |

In terms of image quality, it is shown that the proposed AMBTC is superior to the traditional CAMBTC in the case of PSNR, SSIM, and CD. It is shown that the proposed color AMBTC represents the original image more faithfully than the existing method. In other words, it may know that using three quantization levels is good in the aspect of image quality rather than that of using two quantization levels. Meanwhile, it shows that the CD reflects most of the SSIM and PSNR measurements. However, some results of CD show that the SSIM and PSNR values do not have a proportionally full relationship. For example, CDs of Baboon images are 29.7421 and 26.1272, which have a large gap, but in terms of SSIM and PSNR, the gap is not large. It is reflecting another aspect of the human visual system that the existing PSNR and SSIM cannot measure.

Figure 7 visually compares the Lena image generated from the traditional CAMBTC and the color AMBTC method we proposed. When the images (b) and (c) are enlarged, it can be seen that the (c) image has a clearer hat brim than the (b) image. In terms of image quality, it is natural that the lower the compression ratio, the better the image quality. However, our proposed method improved such limitations.

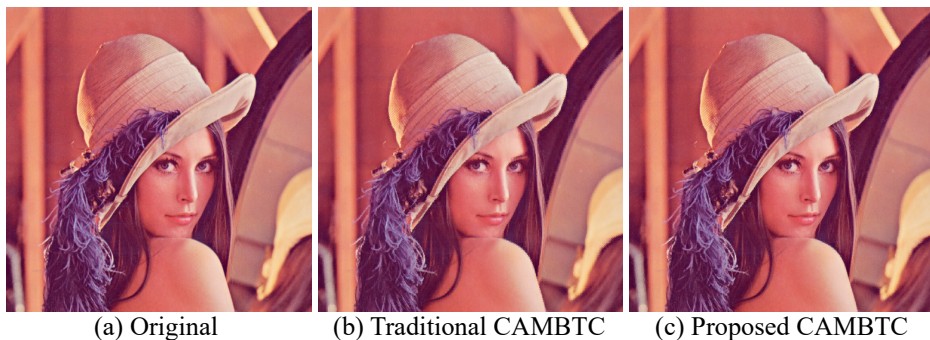

(a) Original     (b) Traditional CAMBTC     (c) Proposed CAMBTC

**Figure 7.** Comparison of original Lena, traditional CAMBTC, and the proposed CAMBTC's images.

In Table 2, EC denotes embedding capacity (bits) for the threshold $T$. Naturally, as the value of $T$ increases, the number of bits increases, and the CD increases. Here, $O$ denotes an original image, and $S$ denotes a stego(marked) image contained data. The difference between PROPOSE #1 and #2 is whether 16 bits are hidden or 32 bits are hidden in the common bitmap $M$, i.e., the first case is a method of concealing 1 bit in each pixel of the bitmap $M$, and the second case is a method of directly concealing 2 bits in each pixel of the common bitmap $M$. In the given three methods, it can be known that when $T$ increases, EC and CD increase simultaneously. PROPOSE#1 has the lowest CD until $T$ is 10. On the other hand, $T$ increases, it is PROPOSE#2 that provides the highest EC. CAMBTC

shows weak performance in terms of EC or CD. The performance of the methods is derived from the difference between the quality of the initial cover image and the EC of the marked image. Table 2 shows the differences. The quality of the initial cover image must be good enough to keep the quality when using the appropriate DH method afterward.

**Table 2.** Comparison of CD between the original image and marked image according to the threshold *T*.

| CAMBTC | $CD_{t(O-M)}$ | T = 5 | | T = 10 | | T = 15 | | T = 20 | |
|---|---|---|---|---|---|---|---|---|---|
| | | EC | $CD_{t(O-S)}$ | EC | $CD_{t(O-S)}$ | EC | $CD_{t(O-S)}$ | EC | $CD_{t(O-S)}$ |
| Lena | 11.5419 | 239,633 | 12.2081 | 438,225 | 14.7086 | 504,081 | 16.0868 | 536,273 | 17.0192 |
| Pepper | 12.4374 | 170,193 | 12.9032 | 384,689 | 15.4278 | 500,129 | 17.6682 | 539,313 | 18.7706 |
| Airplane | 10.3398 | 365,105 | 11.6417 | 460,449 | 12.7021 | 503,825 | 13.5473 | 529,665 | 14.278 |
| Lake | 18.4897 | 180,113 | 18.9001 | 308,353 | 20.3873 | 421,121 | 22.4377 | 475,057 | 23.7682 |
| Baboon | 29.7421 | 129,521 | 29.8709 | 162,593 | 30.194 | 214,673 | 31.1826 | 276,049 | 32.8615 |
| **PROPOSED#1** | $CD_{p(O-M)}$ | T = 5 | | T = 10 | | T = 15 | | T = 20 | |
| | | EC | $CD_{p(O-S)}$ | EC | $CD_{p(O-S)}$ | EC | $CD_{p(O-S)}$ | EC | $CD_{p(O-S)}$ |
| Lena | 10.6977 | 347,489 | 11.3919 | 441,761 | 13.8218 | 472,465 | 15.1749 | 497,649 | 16.8222 |
| Pepper | 12.2976 | 355,665 | 13.679 | 446,321 | 16.3835 | 478,129 | 17.8738 | 500,881 | 19.3211 |
| Airplane | 8.5637 | 406,497 | 9.6877 | 462,929 | 11.2327 | 478,481 | 12.0022 | 493,025 | 13.0615 |
| Lake | 17.0456 | 329,729 | 17.5542 | 389,841 | 19.1579 | 419,105 | 20.4192 | 449,425 | 22.1857 |
| Baboon | 26.1272 | 302,065 | 26.3071 | 331,313 | 27.4723 | 351,073 | 28.6118 | 377,009 | 30.574 |
| **PROPOSED#2** | $CD_{p(O-M)}$ | T = 5 | | T = 10 | | T = 15 | | T = 20 | |
| | | EC | $CD_{p(O-S)}$ | EC | $CD_{p(O-S)}$ | EC | $CD_{p(O-S)}$ | EC | $CD_{p(O-S)}$ |
| Lena | 10.6977 | 399,585 | 12.2793 | 589,025 | 17.0417 | 649,889 | 19.5795 | 698,785 | 22.3666 |
| Pepper | 12.2976 | 416,193 | 14.9094 | 600,257 | 19.9398 | 659,201 | 22.1908 | 704,801 | 24.7835 |
| Airplane | 8.5637 | 517,921 | 10.608 | 631,169 | 13.157 | 661,921 | 14.4979 | 691,585 | 16.2101 |
| Lake | 17.0456 | 365,505 | 18.0093 | 484,577 | 20.953 | 542,145 | 23.1115 | 602,465 | 25.9195 |
| Baboon | 26.1272 | 308,257 | 26.5225 | 365,057 | 28.6964 | 404,353 | 30.6984 | 457,153 | 34.1375 |

Notes: PROPOSE#1 (3Q/B: OPAP, M: 16 bits LSB flipping), PROPOSED#2 (3Q: OPAP, M: 32 bits LSB flipping).

Figure 8 shows the relationship between CD and hidden bits measured using the three DH methods for four images. For Lena (a), PROPOSE#1 is better than the other two methods until bits are 425,000. Also, PROPOSE #1 and #2 show superior performance in terms of CD than CAMABC in the entire x-axis. In Pepper (b), there is a little difference in CD in three methods up to the initial 125,000 bits, and then CD is increased in the range, *x*-axis: 125,000∼300,000. The CD is decreased in the range, *x*-axis: 300,000∼425,000. Also in (b), the CD of PROPOSE # 2 appears to outperform CAMBTC in the entire *x*-axis range.

In Airplane (c), the difference in performance between the CAMABC method and the proposed methods is very large, and both the proposed methods show higher performance than CAMBTC in all ranges. However, PROPOSE#1 is limited embedding ability to 425,000 bits. The Lake (d) also shows a performance graph like that of (c). PROPOSE#1 has a lower EC than that of PROPOSE#2, so after a certain range, its performance degrades sharply. On the other hand, PROPOSE#2 hides 32 bits per block, so initially, the image quality is lower than that of PROPOSE#1, but the moment it exceeds a certain EC, it exceeds the performance limit of PROPOSE#1.

Figure 9 evaluates the quality of Lena marked image created the traditional CAMBTC, PROPOSE#1, and PROPOSE#2 by using the human visual system. The marked images of (a), (b), and (c) embed the same 50,000 bits. The CDs obtained from (a), (b), and (c) are 11.58, 10.7388 and 10.8034, respectively, and in the case of (a), the shoulder line and the right edge of the hat are not smooth.

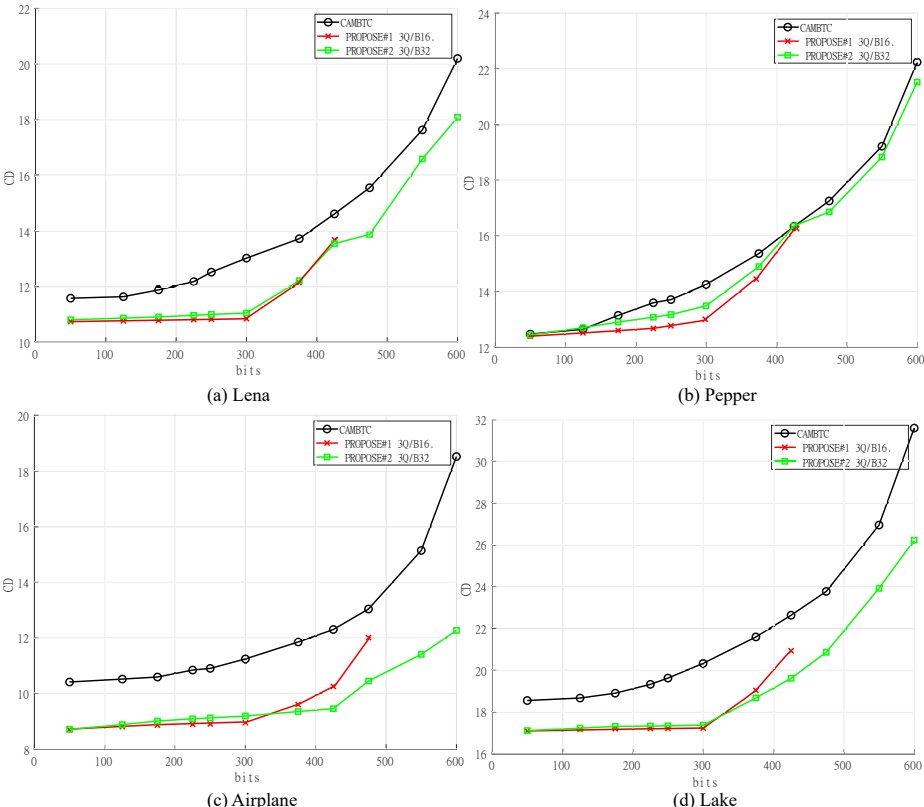

**Figure 8.** CD comparison according to EC.

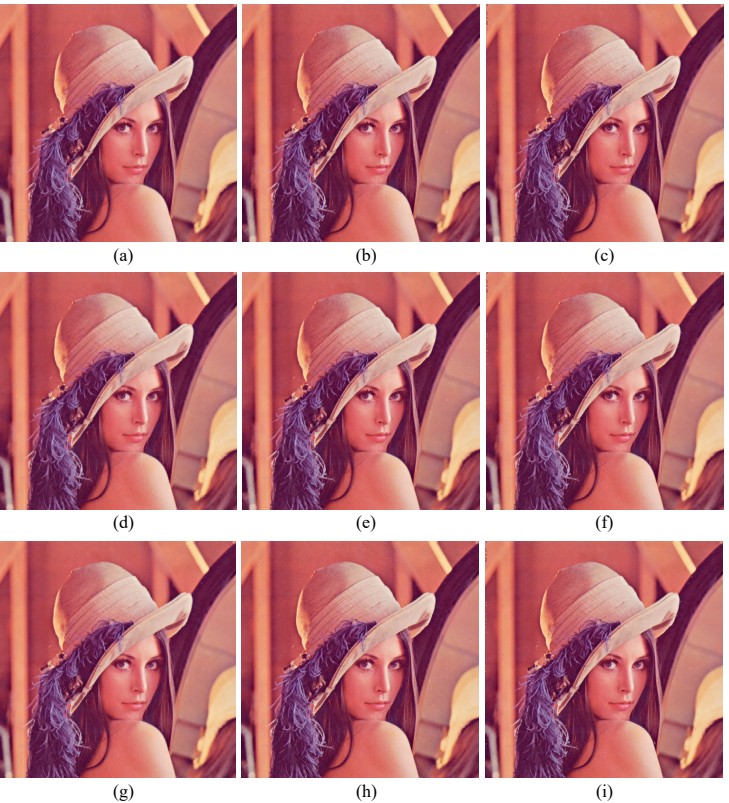

**Figure 9.** Comparisons of Lena image by three methods; (**a**) CAMBTC, EC = 50,000, (**b**) PRO-POSE#1, EC = 50,000, (**c**) PROPOSED#2, EC = 50,000, (**d**) CAMBTC, EC = 250,000, (**e**) PRO-POSE#1, EC = 250,000, (**f**) PROPOSE#2, EC = 250,000, (**g**) CAMBTC, EC = 375,000, (**h**) PROPOSE#1, EC = 375,000, and (**i**) PROPOSE#2, EC = 375,000.

The marked images of (d), (e), and (f) are embedded 250,000 bits and their CDs are measured to be 12.5147, 10.821, and 10.995, respectively. The CD of CAMBTC has about 1.7 difference compared to that of PROPOSED#1. It seems the reason may be related to the quality of the initial compressed image. In the image (d), the texture of the upper right border of the hat is rougher than before. In the case of (f), although the texture of the left boundary is partially roughened, the overall image quality shows similar performance to that of (e).

The images of (g), (h), and (i) have the same EC (=375000 bits), and their CDs are 13.7187, 12.2084, and 12.3063, respectively. In other words, the CD of the image (h) is the best and image(g) is the worst. According to the observation of the human visual system, it was confirmed that a stepwise occurred in the left frame of Lena in (g). On the other hand, (h) and (i) do not seem to have such a problem. There is also noise in the left corner (i). However, other parts of the image are spotless.

## 6. Conclusions

In this paper, we presented a color AMBTC compression method using a common bitmap and k-means, and introduced a method of embedding a secret data into the AMBTC using OPAP. Meanwhile, since the traditional CAMBTC method was derived from compressing a grayscale image, the image is divided into blocks, the RGB layer is separated, and then the AMBTC is applied. Finally, we may obtain compressed two quantization levels and a bitmap per block for each R, G, and B layers. In addition,, the compression ratio of traditional color AMBTC is 96 bits per block. On the other hand, we compressed each block into a common bit-plane and 9 quantization levels, and then further compressed the nine quantization levels to seven bits each. In other words, our proposed color AMBTC requires only 95 bits per block to compress an image. For DH, we proved the fact that the quality of the proposed CAMBTC is superior to the traditional CAMBTC through the simulations. As for the image quality, our suggested method, PROPOSE#1, was evaluated as good, and PROPOSE#2 was evaluated as having the best DH performance. We would like to propose ways to improve the compression performance and quality of AMBTC for color images in the future.

**Author Contributions:** Conceptualization, C.Y. and C.K.; methodology, C.Y. and C.K.; software, D.S.; validation, C.K., L.L. and D.S.; formal analysis, C.Y.; investigation, L.L.; resources, D.S.; data curation, D.S.; writing—original draft preparation, C.K.; writing—review and editing, C.K.; visualization, D.S.; supervision, C.Y.; project administration, C.K.; funding acquisition, L.L. All authors have read and agreed to the published version of the manuscript.

**Funding:** This work was partially supported by the Ministry of Science and Technology (MOST),under Grant 108-2221-E-259-009-MY2 and 109-2221-E-259-010 and by the Basic Science Research Program through the National Research Foundation of Korea (NRF) funded by 2018R1D1A1B07047395 and was supported under the framework of the international cooperation program managed by NRF (2016K2A9A2A05005255) the faculty research fund of Sejong University in 2020. This work was supported by the National Natural Science Foundation of China under Grant 61866028.

**Institutional Review Board Statement:** Not applicable.

**Informed Consent Statement:** Not applicable.

**Data Availability Statement:** Not applicable.

**Acknowledgments:** Thank you to the reviewers who reviewed this paper and the MDPI editor who edited it professionally.

**Conflicts of Interest:** The authors declare no conflict of interest.

**Abbreviations**

The following abbreviations are used in this manuscript:

| | |
|---|---|
| $\mathcal{OI}$ | Original Image |
| $\mathcal{MI}$ | AMBTC Marked Image |
| $\mathcal{P}$ | A Block of an AMBTC |
| $M$ | A block of common bitmap |
| $\mathcal{Q}$ | A Quantization Level |
| LSB | Least Significant Bit |
| DCT | Discrete Cosine Transform |
| BTC | Block Truncation Coding |
| AMBTC | Absolute Moment BTC |
| DH | Data Hiding |
| OPAP | Optimal Pixel Adjustment Process |

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
