# Peer review of "Data Hiding Method for Color AMBTC Compressed Images Using Color Difference"

_applsci, doi:10.3390/app11083418_

Round 1

Reviewer 1 Report

The article is interesting.
The formula of the presented content allows for further research related to image compression and the selection of advanced BITMAP structures in the full spectrum with erroneous records. Very interesting research may lead to the deepening of the research method, among others in the field of analysis of handwriting forgeries with the use of raster and vector graphics.
Yours sincerely

Author Response

We appreciate the reviewer’s valuable comments. The followings are our point-by-point responses:

  • The formula of the presented content allows for further research related to image compression and the selection of advanced BITMAP structures in the full spectrum with erroneous records. Very interesting research may lead to the deepening of the research method, among others in the field of analysis of handwriting forgeries with the use of raster and vector graphics. Yours sincerely.

Response:

Thank you very much for your good evaluation.

Reviewer 2 Report

Excellent detail for an innovative and extremely significant area of research. The only recommendation I can make are minor changes to grammar, syntax and some diction.

Author Response

We appreciate the reviewer’s valuable comments. The followings are our point-by-point responses:

  • Excellent detail for an innovative and extremely significant area of research. The only recommendations I can make are minor changes to grammar, syntax, and some diction.

Response:

Thanks for the appropriate comment. We corrected awkward expressions or misused words in the paper. You can see the revised text, which is blue on the paper.
